# Assessment of Dental Fear and Anxiety Tools for Children: A Review

**DOI:** 10.3390/healthcare13202597

**Published:** 2025-10-15

**Authors:** Mohammed Barry, Mustafa Alnami, Yazeed Thamer Alshobaili, Osama M. Felemban, Heba Jafar Sabbagh

**Affiliations:** 1Pediatric Dentistry Department, Faculty of Dentistry, King Abdulaziz University, Jeddah 21589, Saudi Arabia; mbarry@stu.kau.edu.sa (M.B.); muabalnami@moh.gov.sa (M.A.); yalshobaili@stu.kau.edu.sa (Y.T.A.); omfelemban@kau.edu.sa (O.M.F.); 2Ministry of Health, Specialized Dental Center, Madinah 42315, Saudi Arabia; 3Ministry of Health, Specialized Dental Center, Abha 62562, Saudi Arabia; 4Ministry of Health, Specialized Dental Center, Riyadh 11176, Saudi Arabia

**Keywords:** dental, fear, anxiety, tool, children, pediatric

## Abstract

**Background and Objectives:** Accurate assessment of Dental fear and anxiety (DFA) is crucial for effective management, yet the vast array of measurement tools presents a dilemma for clinicians and researchers in selecting appropriate methods. This literature review aims to provide a comprehensive review and comparison of the commonly used DFA measurement tools in the pediatric dentistry literature. **Methods:** A Comprehensive literature search was conducted in December 2024 using the electronic databases PubMed, MEDLINE, ScienceDirect, and Google Scholar. The search was limited to studies focused on identifying research relevant to DFA in children. For each DFA tool, information on its structure, validity, reliability, strengths, limitations, and target population was recruited and tabulated. A comparison between the DFA tools was then conducted. Quality assessment was performed using Patient-Reported Outcome Measures (PRO). **Results:** The search identified 15 subjective and 5 objective tools. Subjective tools included self-reported scales and pictorial analogs, while objective measures involved physiological monitoring and behavioral observation. While subjective tools offer valuable insights into a child’s self-perceived anxiety, their applicability is influenced by age and cognitive development. Objective measures provide quantifiable data, but require specialized equipment and trained observers. Tools combining simplicity, visual aids, and robust validation were found to be most practical for clinical use. **Conclusions:** A wide range of valid and reliable tools are available to assess DFA in children. Selection should be tailored to the child’s age, cognitive abilities, clinical setting, and clinician training. Combining subjective and objective assessments may enhance diagnostic accuracy.

## 1. Introduction

Dental fear and anxiety (DFA) is a typical emotional response in dental settings, affecting individuals across all age groups. In many cases, anxiety experienced during adolescence or adulthood originates from childhood experiences [1]. Therefore, early diagnosis, prevention, and treatment of DFA in children may contribute to better long-term mental health outcomes.

Globally, the prevalence of DFA in children is estimated to range between 20% and 30% [2], and is influenced by factors such as age, cultural background, socioeconomic status, and environmental exposures [3]. Several risk factors have been identified in the etiology of DFA in children, and these are often shaped by dynamic variables such as age, culture, environmental exposures, and generational shifts [4,5,6,7]. These variations highlight the importance of continuous assessment and a thorough understanding of DFA measurement tools [8].

DFA poses significant challenges to the effective delivery of dental care [9]. It often leads to avoidance behaviors, resulting in poor oral health, increased dental neglect, and a higher likelihood of emergency dental treatments [10,11]. Additionally, DFA can have a negative impact on a child’s self-esteem and overall quality of life [12]. Accurate assessment of DFA is, therefore, critical; early identification allows for timely intervention and the prevention of long-term psychological and physical consequences [13]. Nevertheless, pediatric dentists and caregivers must comprehend the factors that lead to subjective dental fear in children, the consequences of untreated anxiety, and the available techniques for assessing and dealing with it [4,5,7,14].

Assessment tools for DFA vary widely. Some of them are subjective tools, determined by the respondent’s viewpoint, while objective tools depend on observed behaviors and the examiner’s interpretation. Subjective dental fear, unlike objective measurement, relies on the child’s perspective and emotional reaction, rather than directly observable or measurable factors, such as physiological responses [15]. Additionally, whosoever responds to the assessment, whether the child, parent, dentist, or another observer, can significantly impact the results [16,17,18,19].

Pediatric dentists must assess a child’s mental health status and level of DFA before initiating treatment. This includes being familiar with the DFA measurement tools. Although there is a systematic review that compares the outcomes of different DFA tools [20] and a review that assesses studies using these tools [21], there is a shortage of reviews that specifically describe the DFA tools used to evaluate pediatric patient fear and anxiety in the dental clinics in a comprehensive and organized manner.

Therefore, this study aimed to provide a review, evaluation, and comparison of the commonly used DFA measurement scales and tools in the literature of pediatric dentistry, in order to guide clinicians and researchers in selecting appropriate tools for assessing and managing DFA effectively.

## 2. Materials and Methods

A comprehensive literature search was conducted to identify all tools used to assess dental fear and anxiety (DFA) in children, including both subjective and objective measures. In addition, studies evaluating the validity and reliability of DFA tools were included. The search was performed in December 2024 using the electronic databases PubMed, MEDLINE, ScienceDirect, and Google Scholar. The following keywords and Boolean operators were used to retrieve relevant articles: (“dental fear” OR “dental anxiety”) AND (“children” OR “child”) AND (“anxiety scale” OR “anxiety tools”).

The search inclusion criteria included the following: 1—The recruited instrument should be designed specifically for children or have been validated for use in children. 2—The study exclusively assesses dental fear and anxiety. The exclusion criteria were 1—adapted from different tools, 2—tools designed for adults and not used for children, and 3—tools that originally assessed other than DFA, such as pain, and were used in a study to assess DFA.

Search results were screened stepwise: titles, abstracts, and finally, full manuscripts, based on the study aims and eligibility criteria. Eligible studies were reviewed, and data regarding each DFA tool was extracted. Each DFA tool was summarized and tabulated in terms of its definition, validity, reliability, target population, and scoring methods.

A comparative evaluation of DFA tools was conducted, and the strengths and limitations of the subjective tools and DFA scoring items were assessed using the Francis DO et al. It included a Patient Reported Outcome Measures (PRO) checklist, where we used conceptual model (three-items), Content Validity (three-items), Reliability (three-items), Construct Validity (three-items), Scoring & Interpretation (three-items), and Respondent Burden & Presentation (three-items), Ref. [22]; therefore, the scoring will be out of eighteen.

The strengths and limitations of objective physiological measurement tools were evaluated based on Lu JK. et al. 2024 [23] four criteria where adopted from the original paper: continuous monitoring capability (where it will be ideal if the monitoring was constant), device availability and suitability (it will be suitable if it was non-invasive), feasibility of use (it will be ideal if we can use it in research), and cost evaluation (it will be suitable if it come in a low cost).

The search and data extraction was conducted independently by three evaluators (MB, YS, MA). Any disagreements were resolved through discussion with experts (HS and OF).

## 3. Results

Initially, 40 DFA assessment tools were identified from the search results. Eighteen instruments were selected for comprehensive assessment and inclusion in the study, following a screening for relevance and alignment with the aims of this review. Two scales were excluded because they aimed to measure pain in children, eighteen because it assessed DFA in adults, and two because they evaluated negative beliefs towards dentists. The tools are first classified into subjective and objective measurements, and then into their essential characteristics, validity, reliability, and applicability in pediatric dental settings.

### 3.1. Subjective Measurements of DFA

Subjective dental fear pertains to the internal feeling of worry or anxiety that a child has before or during a dental appointment. Subjective dental fear, unlike objective measurements, relies on the child’s perspective and emotional reaction, rather than directly observable or measurable factors, such as physiological responses. Pediatric dentists and caregivers must understand the factors that lead to subjective dental fear in children, the consequences of untreated anxiety, and the techniques that are available for assessing and dealing with it.

#### 3.1.1. Corah Dental Anxiety Scale (C-DAS)

**Definition:** The C-DAS is a self-report scale designed to measure a patient’s level of anxiety about visiting and being at the dentist’s office [24].

**Developed by** Corah in 1969 [25].

**Validity and reliability:** C-DAS demonstrated a stability score of 0.77 when tested twice within one week. It also showed significant correlations with other anxiety measures, including the Dental Anxiety Question (DAQ) (r = 0.77, indicating a strong positive relationship), and the State Trait Anxiety Inventory for Children (STAIC) (r = 0.40–0.58, indicating weak to moderate positive relationships (*p* < 0.01), showing statistical significance [26].

**Target population:** While C-DAS is primarily designed for adults, some studies have used it for children aged 8 years and above [26,27,28].

**Scoring:** It consists of four questions with five answers (a 5-point scale from 1 to 5) for each question [29]. The overall dental anxiety score is derived by totaling the scores of the four questions (ranging from 4 to 20); the higher the score, the higher the level of dental anxiety (Table 1).

#### 3.1.2. Modified Dental Anxiety Scale (MDAS)

**Tool Definition**: The MDAS was introduced as a modified version of the C-DAS. It is a commonly used and validated instrument with which to evaluate dental anxiety in patients.

**Developed by** Humphris et al. in 1995 [30].

**Tool validation and reliability**:

The MDAS has demonstrated internal consistency in children, ranging from α = 0.799 to 0.875 across samples [31,32,33]. Criterion validity was supported by correlations with the Children’s Fear Survey Schedule Dental Subscale (CFSS-DS) and a single fear question (r = 0.44–0.58). Construct validity was supported, as children reported lower anxiety during orthodontic treatment compared to higher anxiety when treatment involved local anesthesia [33]. It has also been validated in Arabic, Spanish, Greek, Chinese, Turkish, Romanian, and Tamil [34,35,36,37,38,39,40,41].

**Target population:** While MDAS is primarily designed for adults, some studies have included children aged 8 years and above [18,42].

**Scoring:** The MDAS differs from the original DAS by adding an extra question that enquires about the patient’s anxiety just before receiving a local anesthetic injection and secondly, by standardizing the response options across all questions, using a consistent 5-point Likert scale to measure the level of anxiety: not anxious (1), slightly anxious (2), fairly anxious (3), very anxious (4), and extremely anxious (5). Similar to the DAS, the total score is calculated by totaling the scores, resulting in a range from 5 to 25, with higher scores indicating greater anxiety [30] (Table 2).

#### 3.1.3. Modified Child Dental Anxiety Scale (MCDAS)

**Tool Definition:** MCDAS is a modified scale of C-DAS, designed to offer an improved degree of sensitivity and precision in assessing dental anxiety among children [18].

**Developed by** Wong et al. in 1998 [43].

**Tool validation and reliability:** The MCDAS demonstrated internal consistency, with item-total correlations ranging from r = 0.60 to 0.74, and Cronbach’s alpha values comparable to those of the Corah Dental Anxiety Scale (C-DAS) and the Children’s Fear Survey Schedule-Dental Subscale (CFSS-DS). Test–retest reliability demonstrated high stability over one week. Validity was supported by strong correlations with the C-DAS and Dental Fear Schedule Subscale-Short Form (DFSS-SF) (r > 0.70), while construct validity was indicated by higher scores in girls (mean = 20.8) compared to boys (mean = 15.6) and by greater anxiety in older children compared to younger ones. The MCDAS was validated for children aged 8 to 15 years [43]. It has also been validated in Italian and Nepali [44,45].

**Target population:** The MCDAS is a reliable and valid self-reported scale for assessing dental anxiety in children aged 8 to 15 years [42,43].

**Scoring:** The scale comprises eight questions designed to assess dental anxiety related to specific dental procedures. Each question is to be answered with a five-point Likert scale, which includes scores ranging from relaxed/not worried (1), slightly worried (2), fairly worried (3), more worried (4), and very worried (5). The total score ranges from 8 to 40, with Higher scores indicating greater dental anxiety (Table 3).

#### 3.1.4. The Faces Version of the Modified Child Dental Anxiety Scale (MCDASf)

**Tool Definition:** It is developed as a modification to the MCDAS by adding a faces analog scale anchored above the original numeric form.

**Developed by** Howard et al. in 2007 [18].

**Tool validation and reliability:** The MCDASf in children aged 5 to 12 years demonstrated internal consistency (Cronbach’s α = 0.82), test–retest reliability (intraclass correlation coefficient = 0.80), and criterion validity through correlation with the Children’s Fear Survey Schedule-Dental Subscale (r = 0.80). Construct validity was supported as children evaluated for dental anxiety, with caries, or requiring dental general anesthesia scored higher than those without these conditions. The MCDASf also showed 51% sensitivity and 79% specificity [18]. It has also been validated in Arabic, Turkish, Croatian, Iranian, Malay, and Chinese [46,47,48,49,50,51].

**Target population:** The MCDASf is a simple, valid, and reliable tool for assessing dental anxiety in young and anxious older children. While Howard et al. validated the MCDASf for children aged 8 to 12 years, they also noted that the scale is suitable for younger children, including those as young as 5 years [18].

**Scoring:** The questions and scoring are similar to MCDAS (Table 4).

#### 3.1.5. Venham Picture Scale (VPS)

**Tool Definition:** It is a projective, non-verbal tool designed to assess anxiety levels in children, particularly in dental settings [52].

**Developed by** Venham and Gaulin-Kremer in 1979 [52].

**Tool validation and reliability:** The VPS in children aged 3 to 8 years demonstrated internal consistency (Kuder–Richardson 20 = 0.838), test–retest reliability over one week (r = 0.70), and construct validity through an inverse correlation with age (r = −0.47) [19]. According to Buchanan and Niven (2002) [53], the Venham Picture Scale (VPS) in children aged 3–18 years showed low mean scores (1.4). It demonstrated concurrent validity through correlation with the Facial Image Scale (r = 0.70, *p* < 0.001) [53].

**Target population:** The VPS is beneficial for children between the ages of 3 and 7, who may have difficulty expressing their anxiety verbally [52]. According to Buchanan and Niven (2002) [53], the tool was effective across a wide age range, from 3 to 18 years. It was quick to administer, making it practical for clinical use with very young children [53].

**Scoring:** The scale consists of eight pairs of images, each showing a child in both an anxious and a calm state. Children are asked to select one image from each pair that best represents how they feel. Each anxious image chosen is scored as 1, while calm selections are scored as 0. The total anxiety score is calculated by totaling the individual scores, with higher totals indicating greater anxiety. The VPS is quick and easy to administer, and it can be used before, during, or after treatment so as to assess a child’s anxiety level [52] (Figure 1).

#### 3.1.6. Children’s Fear Survey Schedule-Dental Subscale (CFSS-DS)

**Tool Definition:** It is a widely used tool designed to assess dental fear and anxiety in children [54]. This subscale is part of the larger Children’s Fear Survey Schedule, which evaluates various fears in children.

**Developed by** Cuthbert and Melamed in 1982 [54].

**Tool validation and reliability:** The CFSS-DS showed high test–retest reliability (r = 0.97), criterion validity with the Kisling and Krebs behavior rating scale (r = −0.62), and construct validity with higher scores in fearful children (mean 37.8–38.1) compared to non-fearful children (mean 20.9–25.0) [54,55]. It is also validated in Arabic, Chinese, Dutch, Finnish, Japanese, Greek, Hindi, Bosnian, Italian, and Swedish [45,55,56,57,58,59,60,61,62,63].

**Target population:** The CFSS-DS is commonly applied to children between the ages of 4 and 14 years, as shown in large population-based studies reviewed by Klingberg and Broberg (2007) [64].

**Scoring:** The scale consists of 15 items describing different dental-related situations or stimuli. Children are asked to rate their fear of each item using a 5-point Likert scale, where 1 represents “not afraid” and 5 represents “very afraid.” The items include: dentists, doctors, injections, having somebody examine your mouth, having to open your mouth, having a stranger touch you, having somebody look at you, the dentist drilling, the sight and noise of the dentist drilling, instruments in the mouth, choking, having to go to the hospital, people in white uniforms, and having the dentist clean your teeth. The total score is calculated by totaling the responses for all 15 items, which range from 15 to 75. Scores from 15 to 31 indicate low dental anxiety, 32 to 38 suggest moderate dental anxiety, and scores of 39 or above reflect high dental anxiety [42,65] (Table 5).

#### 3.1.7. Dental Fear Schedule Subscale-Short Form (DFSS-SF)

**Tool Definition:** It is a shortened version of the original Dental Fear Schedule (DFS), designed to provide a faster, yet reliable and valid method of measuring dental fear [42,66].

**Developed by** Foláyan et al. (2003) [67].

**Tool validation and reliability:** The DFSS-SF was validated for children aged 8 to 13 years, demonstrating internal consistency (α = 0.82) and test–retest reliability (r = 0.73). Criterion validity was supported by a correlation with the Frankl Behavior Rating Scale (r = −0.54), with higher scores in anxious children (mean = 23) compared to non-anxious children (mean = 18.7) [67].

**Target population:** The DFSS-SF is specifically used to assess dental anxiety in children [66]. It was validated by Folayan et al. (2003) and was used mainly with children aged 8 to 13 years [67].

**Scoring:** It evaluates fear levels across various dental situations through a 5-point Likert scale, where 1 represents “not afraid” and 5 represents “very afraid. The scale includes 12 items: making a dental appointment, walking into the dental office, sitting in the waiting room, seeing dental instruments, hearing the drill, seeing the drill, feeling the vibration of the drill, seeing the injection needle, feeling the injection needle, smelling dental chemicals, undergoing an oral examination, and overall fear of dental treatment. Each response is scored, and the sum of all item scores indicates the child’s overall level of dental fear; higher scores reflect greater anxiety [66] (Table 6).

#### 3.1.8. Smiley Faces Program (SFP)

**Tool Definition:** is a computerized tool designed to assess dental fear and anxiety (DFA) in children [65,68].

**Developed by** Buchanan in 2005 [68].

**Tool validation and reliability:** The SFP was validated for children aged 6 to 15 years, demonstrating internal consistency (α = 0.80) and test–retest reliability (r = 0.80). Concurrent validity was supported by correlations with the Children’s Fear Survey Schedule-Dental Subscale (CFSS-DS) (r = 0.60) and the Modified Child Dental Anxiety Scale (MCDAS) (r = 0.60). Construct validity was indicated by higher anxiety ratings for invasive procedures (drill and local anesthetic) compared with waiting room and pre-visit situations [68].

**Target population:** Based on the Modified Dental Anxiety Scale (MDAS), the SFP is intended for children aged 6 to 15 years [65,68].

**Scoring:** This interactive tool uses a seven-point animated faces response scale, where children adjust a neutral face to appear happier or sadder according to their anxiety level. The assessment consists of four items, asking children about their feelings toward: having to go to the dentist the next day for treatment, sitting in the waiting room before a dental appointment, having a tooth drilled, and receiving a local anesthetic injection. Children respond on a scale from 1 to 7, where 1 represents “very happy” (no anxiety) and 7 indicates “extremely sad or nauseous from anxiety.” The total score ranges from 4 (no DFA) to 28 (most severe DFA), providing a reliable and valid measure of dental anxiety in children [65,68]. The responses the children gave were: **1** = Very happy (no anxiety), **2** = Slightly happy, **3** = Neutral, **4** = Slightly sad, **5** = Sad, **6** = Very sad, **7** = Extremely sad/nauseous from anxiety.

#### 3.1.9. Revised Smiley Faces Program (SFP-R)

**Tool Definition:** is a computerized tool designed to assess dental fear and anxiety (DFA) in children. It was developed as an improvement on the original Smiley Faces Program (SFP).

**Developed by** Buchanan in 2005 (2010) [69].

**Tool validation and reliability:** The SFP-R was validated for children aged 4 to 11 years, demonstrating internal consistency (α = 0.70) and test–retest reliability (r = 0.80). Concurrent validity was supported by a correlation with the MCDAS (r = 0.60), and construct validity was indicated by higher scores in girls and older children [69].

**Target population:** It is a computerized dental anxiety measure for children aged 4 to 11 years [69].

**Scoring:** The SFP-R includes an additional item about tooth extraction and an updated animated faces response set, based on child input during small-scale pilot studies. The program is designed using interactive graphics to measure dental fear and anxiety (DFA). Scores for the SFP-R range from 5 (no DFA) to 35 (most severe DFA) [42].

#### 3.1.10. Facial Image Scale (FIS)

**Tool Definition:** It is a simple, pictorial self-report measure designed to assess dental anxiety in children.

**Developed by** Buchanan and Niven in 2002 [53].

**Tool validation and reliability:** According to Buchanan and Niven (2002) [53], the Facial Image Scale (FIS) in children aged 3 to 18 years showed low mean scores (2.2) and no significant effects of age or gender. Concurrent validity was supported through correlation with the Venham Picture Scale (VPS) (r = 0.70, *p* < 0.001) [53].

**Target population:** The scale is designed for children and adolescents aged 3 to 18 years [53].

**Scoring:** The FIS consists of five facial images, ranging from a very happy face to a very unhappy face, illustrating different anxiety levels. Each face is assigned a score from 1 (very happy, no anxiety) to 5 (very unhappy, extreme anxiety). The total score provides an indication of dental anxiety, with higher scores reflecting greater fear [53] (Figure 2).

#### 3.1.11. Abeer Children’s Dental Anxiety Scale (ACDAS)

**Tool Definition:** It is a scale designed to measure dental anxiety in children.

**Developed by** Al-Namankany et al. in 2012 [70].

**Tool validation and reliability:** The ACDAS was validated for children aged 6 years and above, showing internal consistency (α = 0.90) and test–retest reliability (κ = 0.88–0.90). Concurrent validity was supported by correlation with the CFSS-DS (r = 0.77), and a cutoff score > 26 indicated anxiety (sensitivity 96%, specificity 66%). It is also validated in Arabic, Turkish, Malay, and Spanish [71,72,73,74].

**Target population:** This scale was specifically designed for children and adolescents aged six and older.

**Scoring:** The questionnaire consisted of three parts. Part A consists of 13 self-reported questions, arranged in logical order, that ask about the child’s feelings when facing dental experiences. Each question used a 3-face response set. Face 1 represented the feeling of a relaxed, not scared, “happy” person. Face 2 represented a neutral/fair feeling of being “OK.” Face 3 represents the anxious feeling of being “scared.” The child was asked to check under the face that best described his or her response to the question, and a number (1, 2, or 3) was assigned accordingly. The range of values was, therefore, from 13 to 39. The scale indicates that the child is anxious if the score is 26 or higher. 2. Part B comprised three self-reported questions that asked about the child’s feelings of shyness regarding the dentist, or the way his/her teeth looked, and worry about losing control at the dentist; afforded a cognitive assessment; and each required “yes” or “no” as a response. 3. Part C consisted of 3 questions for further assessment of the child, as reported by the legal guardian (to report if the child had a previous experience and how the legal guardian expected his/her child to behave prior to the start of the treatment). The third question was for the dentist to report the child’s behavior at the end of the visit; each question required “yes” or “no” as a response [70] (Figure 3).

#### 3.1.12. Children’s Experiences of Dental Anxiety Measure (CEDAM)

**Tool Definition:** is a child-centered questionnaire designed to assess dental fear and anxiety.

**Developed by** Porritt et al. in 2018 [75].

**Tool validation and reliability:** The CEDAM-14 was validated for children aged 9 to 16 years, demonstrating internal consistency (α = 0.88) and test–retest reliability (ICC = 0.98). Rasch analysis confirmed a unidimensional 14-item structure without age or gender bias. Construct validity was supported by higher scores in clinically anxious children, while concurrent validity was demonstrated through a correlation with the Modified Child Dental Anxiety Scale (MCDAS) (Spearman’s r = 0.67). The CEDAM-14 was also responsive to change, detecting reductions in anxiety after CBT intervention (Cohen’s d = 1.39) [75]. It has additionally been validated in Portuguese and Iranian populations [76,77].

**Target population:** The CEDAM is validated for use in children aged 9 to 16 years.

**Scoring:** The CEDAM consists of 14 items, rated on a three-point scale, with total scores ranging from 14 to 42, with higher scores indicating greater dental anxiety. A manual conversion table transforms raw scores into an interval scale, enhancing precision in clinical and research applications [75] (Table 7).

#### 3.1.13. Shortened Children’s Experiences of Dental Anxiety Measure (CEDAM-8)

**Tool Definition:** The CEDAM-8 is a shortened version of the Children’s Experiences of Dental Anxiety Measure (CEDAM-14), designed to enhance feasibility in clinical and research settings.

Shortened CEDAM (CEDAM-8).

**Developed by** Porritt et al. in 2021 [78].

**Tool validation and reliability:** The CEDAM-8 was validated for children aged 9 to 16 years, showing internal consistency (α = 0.86) with no floor or ceiling effects. Criterion validity was supported by correlation with the CEDAM-14 (r = 0.90), while construct validity was supported by correlation with the global dental anxiety item (r = 0.77). The CEDAM-8 was also responsive to change, detecting meaningful improvement [78].

**Scoring:** The CEDAM-8 consists of eight items, each rated on a three-point scale, where 0 = Not at all like me, 1 = A bit like me, and 2 = A lot like me. The total score ranges from 0 to 16, with higher scores indicating greater levels of dental anxiety [78] (Table 8).

### 3.2. Objective Measurements of DFA

Dental patients typically receive recommendations for treatment modalities based on the clinician’s objective assessment, which may involve behavioral management alone, inhalation sedation, intravenous sedation, or general anesthesia [79].

Objective assessments are especially valuable in situations where patients may not provide accurate reports of their anxiety levels due to denial, insufficient self-awareness, or difficulty in communicating.

#### 3.2.1. Venham’s Clinical Anxiety Rating Scale (VCARS)—(Observational Measurement)

**Tool Definition:** The Venham Clinical Anxiety Rating Scale (VCARS) is an objective behavioral measure designed to assess dental anxiety in children undergoing dental treatment [19].

**Developed by** Venham et al. in 1977 [19].

**Tool validation and reliability:** According to Venham et al. (1980), the VCARS was validated in preschool children aged 2 to 5 years, showing high inter-rater reliability (r = 0.78–0.98) and construct validity through significant score differences across sequential visits, and by comparison with physiological measures (heart rate) and the Venham Picture Scale (VPS) [80].

**Target population:** The target population for VARS is generally children aged 3 to 12 years [80].

**Scoring:** VARS is constructed of six behaviorally defined categories that range from 0 to 5, with a higher score indicating a higher amount of anxiety. The score gradually increases from relaxed, smiling, willing, and able to converse (Score 0), to the child is totally out of control (Score 5) (Table 9).

#### 3.2.2. Heart Rate Monitoring (Physiological Measurement)

Increased heart rate is a typical physiological response to anxiety [81]. Measuring heart rate in beats per minute (bmp) during dental visits can offer an objective evaluation of a patient’s level of fear and anxiety. Heart rate monitors and pulse oximeters are used to measure changes in heart rate in response to dental treatment. Elevated heart rate can serve as an indicator of heightened anxiety, thus necessitating the implementation of measures to manage anxiety. It is a safe, non-invasive method that is converted to a numerical score by applying age-specific thresholds. Accurate reference ranges are key to assessing whether a vital sign is abnormal [82]. However, it is not constant, and although in a specific range, it varies between children. Thus, to assess anxiety, two to three measurements are required (before, during, and after the procedure). According to studies, Heart Rate (HR) monitoring was validated in children aged 6 to 12 years, with mean values ranging from 80 to 118 bpm. Validity was supported by correlations with the Corah Dental Anxiety Scale (r = 0.57) and the numeric Visual Facial Anxiety Scale (rs = 0.48) [15,83].

#### 3.2.3. Blood Pressure (Physiological Measurement)

Blood pressure is a useful physiological indicator for evaluating dental anxiety and stress; elevated blood pressure during dental treatment may reflect the patient’s emotional discomfort and anxiety in the clinical environment [84]. Multiple studies found that the increase in blood pressure (BP) correlates with a significant increase in dental anxiety during extraction and local anesthesia [85,86]. The measurement of blood pressure (BP) is usually recorded 5 min before the dental procedure as a baseline, during the dental procedure, and post-procedure to track normalization. According to studies, Blood Pressure (BP) monitoring was validated in children aged 8 to 12 years, with a mean systolic pressure of 125.0 ± 9.4 mmHg and a mean diastolic pressure of 79.9 ± 5.7 mmHg. Validity was supported by a positive correlation between systolic BP and the numeric Visual Facial Anxiety Scale (rs = 0.29), while diastolic BP showed no significant correlation [15].

#### 3.2.4. Salivary Cortisol Levels (Physiological Measurement)

Saliva is widely recognized as a valuable diagnostic tool in healthcare. Due to its non-invasive collection method and diverse composition, it is well-suited for evaluating biomarkers associated with different illnesses [87]. Salivary diagnostics have the potential to detect diseases early, track their progression, and provide individualized treatment options, which could significantly change current healthcare practices [87].

Cortisol, also referred to as the stress hormone, is an essential hormone that is secreted by the adrenal glands in response to stress. It plays a vital role in regulating the body’s stress response and influencing multiple physiological systems. Cortisol levels vary throughout the day, but continuous or extreme stress can result in chronically increased levels, which can have adverse consequences on both physical and mental well-being [88,89]. Salivary Cortisol was validated for children from birth to 14 years of age [87,89]. Salivary cortisol is a reliable indicator of physiological stress due to its association with the Hypothalamic–Pituitary–Adrenal (HPA) axis activity; it is a non-invasive method for children and has the ability to distinguish between children with and without anxiety [90].

The collection methods:

Unsimulated method: This method can be achieved by instructing the individual to allow saliva to accumulate naturally in their mouth before expelling it into a sterile vial or collection tube, ensuring that they avoid swallowing during the process. Alternatively, the Salivette kit instructs the individual to place a sterile cotton swab from the Salivette kit under their tongue or between their gum and cheek without chewing. After approximately two minutes, the cotton swab is removed and placed into the collection tube for subsequent analysis.

The simulated method involves instructing the participant to chew on a sterile cotton swab from the Salivette kit for approximately 1–2 min, stimulating saliva production and saturating the swab. Once adequately saturated, the swab is placed back into the collection tube provided. After saliva collection, the sample should be stored in a sub-zero freezer in order to preserve its integrity until the sample is analyzed using a specialized Enzyme-Linked Immunosorbent Assay (ELISA) [91].

#### 3.2.5. Electrodermal Activity (Physiological Measurement)

Electrodermal Activity (EDA) measures the activation of sweat glands, which are solely under sympathetic nervous system control. It is a highly sensitive and somatically independent indicator of sympathetic arousal, making it a valuable tool for assessing the emotional and cognitive responses of children experiencing dental anxiety [84]. It is usually recorded by using an EDA sensor device, 3–5 min before the procedure, during the procedure, and 3 min after the procedure. Higher reading indicates dental anxiety.

EDA was validated for children aged 5 to 13 years and for children with developmental disabilities [92,93]. It also demonstrated reliable functional consistency across clinical settings, particularly in non-verbal and cognitively impaired populations [92,93,94].

Table 10 presents DFA tools along with their age ranges, questions and answers, scales and scoring, strengths, limitations, and notes on validity, reliability, and validated languages. It also reports the quality score of the DFA tools according to the PRO checklist. Appendix A reports the DFA quality scoring in more detail.

All the dental anxiety tools included a detailed explanation of their purpose and target audience. Each tool was developed by experts, and most of the questionnaires were brief and simple to use. In general, studies demonstrated consistent results when the measures were repeated, thus indicating good reliability. Additionally, the tools’ ability to distinguish between children with high anxiety and those with low anxiety, as well as their expected associations with other anxiety measures, provided evidence of their validity.

Despite these advantages, certain disadvantages were noticed. Except for the CEDAM and the CEDAM-8, the majority of the measures were not made to monitor change over time. The fact that none of the tools provided guidance on how to handle missing answers was another disadvantage (Appendix A). Table 11 provides further details regarding each tool’s validity and reliability.

## 4. Discussion

Assessment tools for Dental Fear and Anxiety (DFA) are important for pediatric dentists to understand and utilize in their daily practice with their patients and in research. Given the numerous DFA tools available in the literature, which vary in complexity, subjectivity, and validation criteria, this review aims to provide a comprehensive summary of these tools in order to support pediatric dentists in their selection and application.

Subjective tools, although generally easier to use, measure anxiety based on the individual’s perception of their symptoms. Factors such as personal experience, cognitive ability, and maturity can influence the results, thereby potentially leading to reporting bias [64,95].

Moreover, the C-DAS and its modified versions, MDAS, offer a brief assessment; however, the need for verbal and cognitive ability limits the use of these tools. In addition, they are both limited in scope, measuring a narrow range of dental situations. The MCDAS and its facial version (MCDASf) are easy to use for children, due to the modification of the questions and the inclusion of faces, and, to a lesser extent, the VPS and FIS. Making them more suitable for younger children.

In contrast, the CFSS-DS and the DFSS-SF provide broader coverage of DFA situations. Still, the need for cognitive ability and the length of the tools remain obstacles to their practical use.

While the SFP and its revised version, SFP-R, are computerized tools that have been developed to enhance the child’s involvement through interactive visual methods, their cost and access to the tools remain a challenge in utilizing them on a broad scale for children, unlike the other tools.

Recent tools have been developed to measure DFA, such as the CEDAM and CEDAM-8, although these tools have demonstrated excellent psychometric properties; however, the length of the scales and the inability to use them with younger children remain significant limitations for the scales. ACDAS overcomes these obstacles by using short, simple questions and incorporating faces so as to facilitate straightforward answers, making it a suitable choice for use in the clinic.

Subjective tools are easy to administer, non-invasive, time-efficient, and offer good validity and reliability. They can be used in different settings with a wide range of children’s ages. Dentists can optimize their use by selecting tools based on the child’s age, literacy level, clinical context, and the purpose of the assessment.

Additionally, multiple studies have evaluated DFA using various tools on the same sample and found no statistically significant differences among them [70,96,97,98,99]. This suggests that their effectiveness in measuring DFA is comparable, allowing dentists flexibility in both clinical application and research. However, studies that utilized DFA tools to assess different behavior management techniques (BMTs) have reported varying outcomes. For example, Bagher et al. compared two types of BMTs and reported similar outcomes related to VCARS, ACDAS, and heart rate, with no significant differences between the techniques. However, salivary cortisol levels showed greater variation between the BMTs [100]. In addition, Khogeer et al. (2025) [101] assessed dental anxiety using FIS, VCARS, and heart rate. They found that changes in dental anxiety across different BMTs were not consistent. The self-reporting tool (FIS) demonstrated greater changes compared to evaluations conducted by an observer, such as the dentist (VCARS). Heart rate changes were greater than those measured by VCARS but less than those measured by FIS [101]. This could also indicate that a child’s self-perception may reflect preference more than actual anxiety. Thus, we recommend using multiple complementary assessment tools in order to achieve a more reliable evaluation of dental anxiety.

To date, there is no universally accepted gold standard for assessing DFA. Consequently, most studies establish criterion validity by comparing outcomes across instruments. In this review, we have summarized the comparisons in Table 11. Therefore, until a gold standard is established, triangulation using multiple validated tools and context-specific validation is recommended.

In addition, multiple DFA tools (CFSS-DS, MDAS, MCDAS, MCDASf, ACDAS, CE-DAM) have been employed across different populations. These tools were translated, validated through face and content validity, and pilot tested to ensure proper cultural adaptation. Moreover, observer-based ratings such as FIS and ACDAS incorporated either culturally neutral or locally relevant imagery and behavioral anchors. Finally, objective physiological measures, such as heart rate, should be interpreted relative to individual resting baselines and culture-specific normative ranges [82].

A notable limitation across all subjective tools is the lack of a strategy for handling missing data, which, if it occurs, could affect the questionnaire and result in measurement bias, jeopardizing the dentist’s decision-making and the choice of a proper treatment plan. It can be overcome by the dentist ensuring that the child answers all the questions and combining clinical judgment with the questionnaire [102].

On the other hand, objective tools do not require the child’s participation in answering questionnaires. It depends on observable symptoms, as examined by clinicians, or on measuring physiological changes [103]. It is particularly valuable for children who are very young or have a cognitive disability.

Venham Clinical Anxiety Rating Scale (VCARS) is a valuable tool for clinicians to use for children of all ages and cognitive states, due to its simplicity and clarity. However, examiners’ calibration and training are vital for using objective clinician evaluation.

Additionally, heart rate is often used to assess a patient’s anxiety by monitoring changes before, during, and after dental treatment. Studies have demonstrated variations in mean heart rate among children subjected to different behavior management techniques, such as Ask-Tell-Ask and Tell-Play-Do, compared to the traditional Tell-Show-Do method [103]. Heart rate is typically used in conjunction with other anxiety measurement approaches to provide a more comprehensive assessment [100,104]. However, a heart rate monitor requires a certain degree of cooperation from the child or assistance from the dental team to be used during the dental visit [105]. Additionally, it is influenced by the child’s level of activity, age, and can vary between genders [106]. It is also affected by the devices.

Moreover, measuring blood pressure is affordable, and the results are standardized, with minimal cooperation from the child. However, studies have shown that blood pressure ranges can vary based on a child’s age, gender, obesity, and geographic location, making it a challenging tool to be used as a reliable indicator for diagnosing anxiety in children [107,108]. Additionally, it lacks continuity of measurement and may be descriptive during dental treatment.

Salivary cortisol is a reasonable measure of anxiety due to its non-invasive method; however, the cost and time required to collect samples before, during, and after, make it impractical to use outside the research setting. Additionally, it has variable ranges and is influenced by multiple variables [106].

EDA is a valuable technique for evaluating DFA in children, where variations in skin conductance levels, influenced by sympathetic nervous system activity, act as physiological markers of anxiety or stress. It is non-invasive and sensitive to acute emotional arousal, making it beneficial for anxiety assessment during dental treatments. Still, it costs more than other tools and could cause discomfort to children during treatment.

While objective tools provide valuable psychological and behavioral data, their practical application is limited by the availability of resources across different settings. Tools like heart rate monitors and blood pressure devices are affordable and available in various clinical settings, unlike salivary cortisol and EDA, which are more challenging to obtain and are mostly used in research settings and specialized environments.

Despite considerable progress in scale development and techniques for identifying anxious children, there remains potential for innovation, particularly with advancements in technology and artificial intelligence that could support the detection and treatment of anxiety. However, the validation and reliability of these tools remain critical areas that require further evaluation and meta-analysis across different populations and age groups so as to have consistently more accurate measurement tools.

## 5. Conclusions

A wide range of valid and reliable tools are available to assess DFA in children. Selection should be tailored to the child’s age, cognitive abilities, clinical setting, and clinician training. This review serves as a practical reference to aid in evidence-based decision-making for DFA management in pediatric dentistry.

## Figures and Tables

**Figure 1 healthcare-13-02597-f001:**
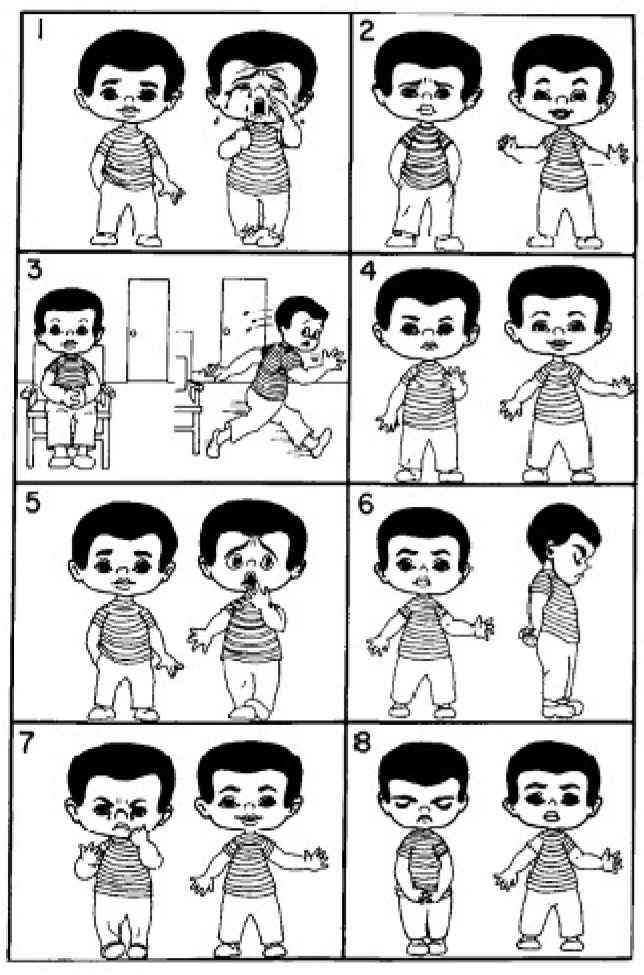
Venham Picture Scale (VPS).

**Figure 2 healthcare-13-02597-f002:**
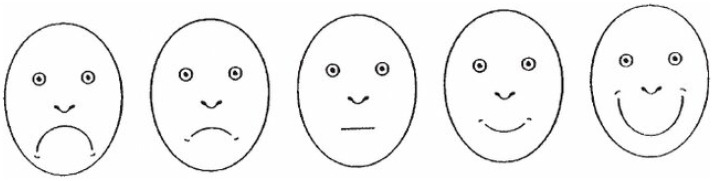
Facial Image Scale (FIS).

**Figure 3 healthcare-13-02597-f003:**
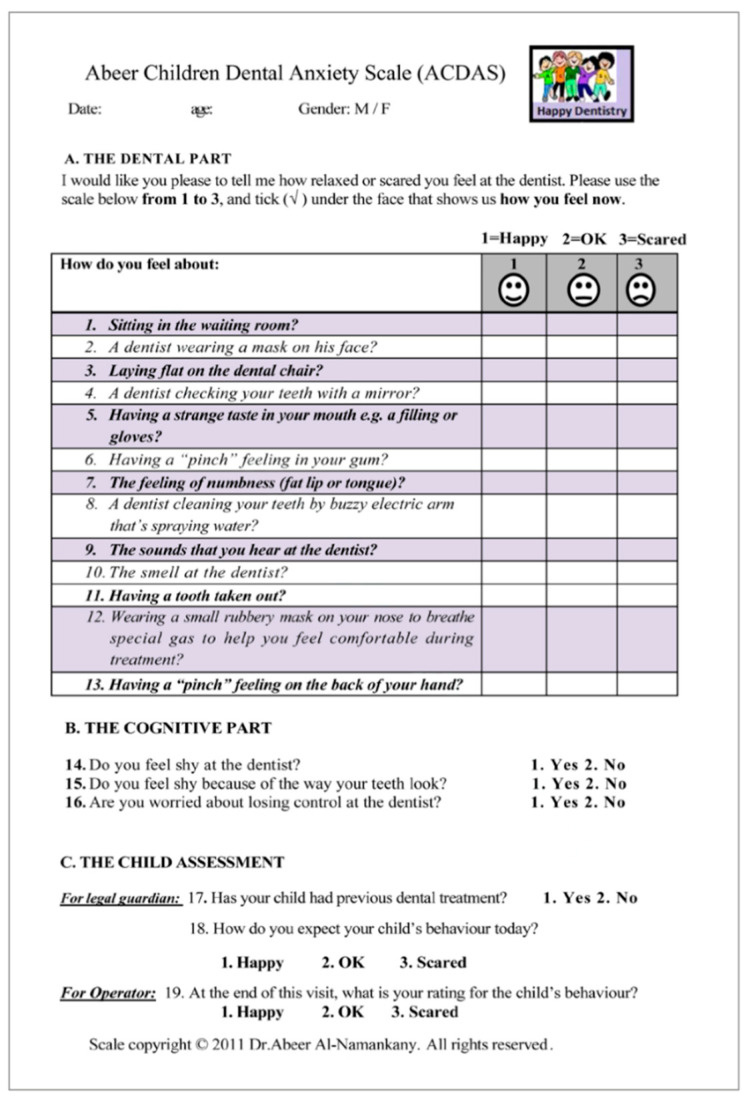
Abeer Children’s Dental Anxiety Scale (ACDAS).

**Table 1 healthcare-13-02597-t001:** Corah Dental Anxiety Scale (C-DAS).

Question	Answer
1. If you had to go to the dentist tomorrow, how would you feel about it?	(a) I would look forward to it as a reasonably enjoyable experience. (b) I wouldn’t care, one way or the other. (c) I would be a little uneasy about it. (d) I would be afraid that it would be unpleasant and painful. (e) I would be very frightened of what the dentist might do.
2. When you are waiting in the dentist’s office for your turn in the chair, how do you feel?	(a) Relaxed. (b) A bit uneasy. (c) Tense. (d) Anxious. (e) So anxious that I sometimes break out in a sweat or almost feel physically sick.
3. When you are in the dentist’s chair waiting, while he gets his drill ready to begin working on your teeth, how do you feel?	(a) Relaxed. (b) A bit uneasy. (c) Tense. (d) Anxious. (e) So anxious that I sometimes break out in a sweat or almost feel physically sick.
4. You are in the dentist’s chair to have your teeth cleaned. While you are waiting and the dentist is setting out the instruments which he will use to scrape your teeth around the gums, how do you feel?	(a) Relaxed. (b) A bit uneasy. (c) Tense. (d) Anxious. (e) So anxious that I sometimes break out in a sweat or almost feel physically sick.

**Table 2 healthcare-13-02597-t002:** Modified Dental Anxiety Scale (MDAS).

Item	Situation	Not Anxious	Slightly Anxious	Fairly Anxious	Very Anxious	Extremely Anxious
1	If you went to your dentist for treatment tomorrow, how would you feel?	☐	☐	☐	☐	☐
2	If you were sitting in the waiting room (waiting for treatment), how would you feel?	☐	☐	☐	☐	☐
3	If you were about to have a tooth drilled, how would you feel?	☐	☐	☐	☐	☐
4	If you were about to have your teeth scaled and polished, how would you feel?	☐	☐	☐	☐	☐
5	If you were about to have a local anesthetic injection in your gum, how would you feel?	☐	☐	☐	☐	☐

**Table 3 healthcare-13-02597-t003:** Modified Child Dental Anxiety Scale (MCDAS).

Questions	Answers
1. How do you generally feel about going to the dentist?	(a) Relaxed/not worried, (b) slightly worried, (c) Fairly worried, (d) More worried (e) Very worried.
2. How do you feel about having your teeth looked at?	(a) Relaxed/not worried, (b) slightly worried, (c) Fairly worried, (d) More worried, (e) Very worried.
3. How do you feel about having your teeth scraped and polished?	(a) Relaxed/not worried, (b) slightly worried, (c) Fairly worried, (d) More worried (e) Very worried.
4. How do you feel about having an injection in the gum?	(a) Relaxed/not worried, (b) slightly worried, (c) Fairly worried, (d) More worried (e) Very worried
5. How do you feel about having a filling?	(a) Relaxed/not worried, (b) slightly worried, (c) Fairly worried, (d) More worried (e) Very worried
6. How do you feel about having a tooth extracted?	(a) Relaxed/not worried, (b) slightly worried, (c) Fairly worried, (d) More worried (e) Very worried
7. How do you feel about being put to sleep in order to have treatment?	(a) Relaxed/not worried, (b) slightly worried, (c) Fairly worried, (d) More worried (e) Very worried
8. How do you feel about having a mixture of “gas and air” which will help you feel comfortable for treatment, but cannot put you to sleep?	(a) Relaxed/not worried, (b) slightly worried, (c) Fairly worried, (d) More worried (e) Very worried

**Table 4 healthcare-13-02597-t004:** The Faces version of the Modified Child Dental Anxiety Scale (MCDASf).

	** 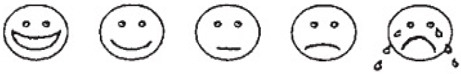 **
Item	(1) Relaxed/Not Worried	(2) Slightly Worried	(3) Fairly Worried	(4) More Worried	(5) Very Worried
generally, going to the dentist?	☐	☐	☐	☐	☐
having your teeth looked at?	☐	☐	☐	☐	☐
having your teeth scraped and polished?	☐	☐	☐	☐	☐
having an injection in the gum?	☐	☐	☐	☐	☐
having a filling?	☐	☐	☐	☐	☐
having a tooth extracted?	☐	☐	☐	☐	☐
being put to sleep in order to have treatment?	☐	☐	☐	☐	☐
having a mixture of ‘gas and air’ which will help you feel comfortable for treatment, but cannot put you to sleep?	☐	☐	☐	☐	☐

**Table 5 healthcare-13-02597-t005:** Children’s Fear Survey Schedule-Dental Subscale (CFSS-DS).

Item	Subject	Not Afraid at All (1)	A Little Afraid (2)	Somewhat Afraid (3)	Afraid (4)	Very Afraid (5)
1	Dentist	1	2	3	4	5
2	Doctor	1	2	3	4	5
3	Injections (shots)	1	2	3	4	5
4	Having somebody examine your mouth	1	2	3	4	5
5	Having to open your mouth	1	2	3	4	5
6	Having a stranger touch you	1	2	3	4	5
7	Having somebody look at you	1	2	3	4	5
8	The dentist drilling	1	2	3	4	5
9	The sight of the dentist drilling	1	2	3	4	5
10	The noise of the dentist drilling	1	2	3	4	5
11	Having somebody put instruments in your mouth	1	2	3	4	5
12	Choking	1	2	3	4	5
13	Having to go to the hospital	1	2	3	4	5
14	People in white uniforms	1	2	3	4	5
15	Having the nurse clean your teeth	1	2	3	4	5

**Table 6 healthcare-13-02597-t006:** Dental Fear Schedule Subscale-Short Form (DFSS-SF).

Item	1 (Not Afraid)	2 (A Bit Afraid)	3 (Fairly Afraid)	4 (Quite Afraid)	5 (Very Afraid)
Making a dental appointment	☐	☐	☐	☐	☐
Walking into the dental office	☐	☐	☐	☐	☐
Sitting in the waiting room	☐	☐	☐	☐	☐
Seeing dental instruments	☐	☐	☐	☐	☐
Hearing the drill	☐	☐	☐	☐	☐
Seeing the drill	☐	☐	☐	☐	☐
Feeling the vibration of the drill	☐	☐	☐	☐	☐
Seeing the injection needle	☐	☐	☐	☐	☐
Feeling the injection needle	☐	☐	☐	☐	☐
Smelling dental chemicals	☐	☐	☐	☐	☐
Undergoing an oral examination	☐	☐	☐	☐	☐
Overall fear of dental treatment	☐	☐	☐	☐	☐

**Table 7 healthcare-13-02597-t007:** Children’s Experiences of Dental Anxiety Measure (CEDAM).

Item	Responses
1. When I know I have an appointment with a dentist…	1. I will do nothing to avoid going 2. I will do some things to avoid going 3. I will do everything to avoid going
2. When I know I have an appointment with a dentist…	1. I will tell my parents/carers that I don’t mind going 2. I will tell my parents/carers that I would rather not go 3. I will tell my parents/carers that I really don’t want to go
3. When I next visit the dentist…	1. I will let the dentist look in my mouth 2. I will try to stop the dentist from looking in my mouth a bit 3. I will not let the dentist look in my mouth
4. When I next visit the dentist…	1. I will not get worried if the dentist tells me I need to have something done 2. I will get a bit worried if the dentist tells me I need to have something done 3. I will get really worried if the dentist tells me I need to have something done
5. When I next visit the dentist…	1. If I asked the dentist to stop what they were doing they would definitely stop 2. If I asked the dentist to stop what they were doing they might stop 3. If I asked the dentist to stop what they were doing they would not stop
6. When I next visit the dentist I think…	1. I will not be worried that it will be painful 2. I will be a little worried that it will be painful 3. I will be very worried that it will be painful
7. When I next visit the dentist I think…	1. Nothing will go wrong 2. Something will go a bit wrong 3. Something will go very wrong
8. When I next visit the dentist I think…	1. I will have a lot of control over what happens in the appointment 2. I will have a bit of control over what happens in the appointment 3. I will not have any control over what happens in the appointment
9. When I next visit the dentist I think I will…	1. Not feel shaky 2. Feel a bit shaky 3. Feel very shaky
10. When I next visit the dentist I think I will…	1. Not feel stressed 2. Feel a bit stressed 3. Feel very stressed
11. When I next visit the dentist I think I will…	1. Not feel upset 2. Feel a bit upset 3. Feel very upset
12. When I next visit the dentist I think I will…	1. Not feel embarrassed 2. Feel a bit embarrassed 3. Feel very embarrassed
13. When I next visit the dentist I think I will…	1. Not feel angry 2. Feel a bit angry 3. Feel very angry
14. When I next visit the dentist I think I will…	1. Feel that I can completely trust the dentist 2. Feel that I can only trust the dentist a bit 3. Feel that I can’t trust the dentist

**Table 8 healthcare-13-02597-t008:** Shortened Children’s Experiences of Dental Anxiety Measure (CEDAM-8).

Item	Responses
1. I get nervous when I think about going to the dentist.	0 = Not at all like me, 1 = A bit like me, and 2 = A lot like me
2. I feel worried before my dental appointment.	0 = Not at all like me, 1 = A bit like me, and 2 = A lot like me
3. I feel shaky when I have to go to the dentist.	0 = Not at all like me, 1 = A bit like me, and 2 = A lot like me
4. I avoid talking about my dental visits.	0 = Not at all like me, 1 = A bit like me, and 2 = A lot like me
5. I try not to think about going to the dentist.	0 = Not at all like me, 1 = A bit like me, and 2 = A lot like me
6. I feel my heart beating faster when I am at the dentist.	0 = Not at all like me, 1 = A bit like me, and 2 = A lot like me
7. I feel sick when I have to go to the dentist.	0 = Not at all like me, 1 = A bit like me, and 2 = A lot like me
8. I find it hard to breathe when I go to the dentist.	0 = Not at all like me, 1 = A bit like me, and 2 = A lot like me

**Table 9 healthcare-13-02597-t009:** Venham Clinical Anxiety Rating Scale (VCARS).

0	Relaxed, smiling, willing, and able to converse
1	Uneasy, concerned. During stressful procedure may protest briefly and quietly to indicate discomfort. Hands remain down or partially raised to signal discomfort. Child willing and able to interpret experience as requested. Tense facial expression, may have tears in eyes
2	Child appears scared. Tone of voice, questions and answers reflect anxiety. During stressful procedure, verbal protest, (quiet) crying, hands tense and raised, (not interfering much, may touch dentist’s hand or instrument, but not pull at it). child interprets situation with reasonable accuracy and continues to work to cope with his/her anxiety
3	Shows reluctance to enter situation, difficulty in correctly assessing situational threat. Pronounced verbal protest, crying. Using hands to try to stop procedure. Protest out of proportion to threat. Copes with situation with great reluctance
4	Anxiety interferes with ability to assess situation. General crying not related to treatment. More prominent body movement. Child can be reached through verbal communication, and eventually with reluctance and great effort he or she begins the work of coping with the threat
5	Child out of contact with the reality of the threat. Common loud crying, unable to listen to verbal communication, makes no effort to cope with threat. Actively involved in escape behavior. Physical restraint required

**Table 10 healthcare-13-02597-t010:** Dental Fear and Anxiety Assessment Tools and Their Characteristics.

Dental/Fear Anxiety Tools	Age Range	Questions/Answer	Scale/Scoring	Strengths ^	Limitations ^	Validity and Reliability +/+ /Language/Quality Score *
Subjective anxiety tools						
1. Corah Dental Anxiety Scale (C-DAS).	Children aged 8-years and older.	4 item/5 multiple choice answers.	1–5 for each item, total 4–20 with higher scores indicating greater anxiety.	Clear construct Simple scoring.	Narrow scope. No MDP	+\+ English. 14
2. Modified Dental Anxiety Scale (MDAS).	Children aged 7-years and older.	5 items/5 Likert-scale answer.	The results range from 5 to 25, with higher scores indicating greater anxiety.	Clear construct, improves C-DAS Simple scoring.	Narrow scope. No MDP	+\+ English, Arabic, Spanish, Greek, Chinese, Turkish, Romanian, and Tamil. 14
3. Modified Child Dental Anxiety Scale (MCDAS).	Children aged 8 to 15 years.	8 items/5 Likert-scale answer.	The results range from 8 to 40 with higher scores indicating greater anxiety.	Child-focused, simple wording, Covers a wide range of dental procedures, Easy scoring.	Not suitable for younger children. No MDP	+\+ English, Italian, Nepali. 15
4. Faces Version of The Modified Child Dental Anxiety Scale (MCDASf).	children aged 5 years and older.	8 items/5 Likert-scale answer, with facial analogs.	The results range from 8 to 40 with higher scores indicating greater anxiety.	Child-friendly visual format, suitable for younger ages.	No MDP	+\+ English, Arabic, Turkish, Croatian, Iranian, Malay, and Chinese. 15
5. Venham Picture Scale (VPS).	Children aged 3 years and older.	Picture pairs.	0 for calm, 1 for anxiety, the higher, the more anxious the child.	Very quick, simple, picture-based measure.	Less sensitive than multi-item Likert scales. No MDP	+\+ Not Applicable. 15
6. Children’s Fear Survey Schedule-Dental Subscale (CFSS-DS).	Children aged 4–14 years.	15 items/5 Likert-scale answer.	The results range from 15 to 75 with higher scores indicating greater anxiety.	Ease of use, Clear scoring, Designed for school-aged childre.	No MDP	+\+ English, Arabic, Chinese, Dutch, Finnish, Japanese, Greek, Hindi, Bosnian, Italian, and Swedish. 15
7. Dental Fear Schedule Subscale-Short Form (DFSS-SF).	Children aged 8–13 years.	12 items/5 Likert-scale answer.	The results range from 12 to 60 with higher scores indicating greater anxiety.	Covers several aspects of dental fear, Provides both total and subscale scores.	Longer than brief tools. No MDP	+\+ English. 15
8. Smiley Faces Program (SFP).	Children aged 6–15 years.	4 items/7 Likert-scale answer.	The results range from 4 to 28 with higher scores indicating greater anxiety.	Child-friendly visual format, quick, simple.	Requires access to a computer. No MDP	+\+ English. 14
9. Revised Smiley Faces Program (SFP-R).	Children aged 4–11 years.	5 items/7 Likert-scale answer.	The results range from 5 to 35 with higher scores indicating greater anxiety.	simple and engaging, face-based format, Suitable for children with limited reading skills.	Requires access to a computer. No MDP	+\+ English. 15
10. Facial Image Scale (FIS).	Children aged 3–18 years.	Facial analogs.	1–5 with higher scores reflecting greater fear.	Very quick and simple tool.	Less sensitive than multi-item Likert scales. No MDP	+\+ Not Applicable. 15
11.Abeer Children Dental Anxiety Scale (ACDAS).	Children aged 6 and older.	13 items/3 Likert-scale answer, with facial analogs.	the child is anxious if the score is ≥26.	Multidimensional Includes cognitive, emotional, and behavioral domains, Clinical utility.	Length, less suitable for younger children. No MDP	+\+ English, Arabic, Turkish, Malay, and Spanish. 15
12. Children’s Experiences of Dental Anxiety Measure (CEDAM).	Children aged 9–16 years.	14 item/3 multiple choice answers.	The results range from 14 to 42 with higher scores indicating greater anxiety.	Quick and feasible, Useful clinically.	Not suitable for younger children. No MDP	+\+ English, Portuguese and Iranian. 16
13. Shortened Children’s Experiences of Dental Anxiety Measure (CEDAM-8).	Children aged 9–16 years.	8 item/3 Likert-scale answers.	0–16 with higher scores indicating greater anxiety.	Short, efficient, quick, Covers multiple aspects of dental anxiety.	Not suitable for younger children, Shortened form may lose some nuance compared to the full CEDAM. No MDP	+\+ English. 16
Objective anxiety tools						
1. Venham’s Clinical Anxiety Rating Scale (VCARS).	Children aged 3–12 years.	Observational; 6-point behavior-based scale (0–5).	0–5 scale, higher scores indicate more anxiety.	Simple, quick, does not require verbal response.	Requires trained observers, subjective interpretations possible.	+\+ English. 14
2. Heart Rate Monitoring.	All ages.	Physiological measure using a pulse oximeter or heart rate monitor.	Percentage or absolute increase in heart rate compared to baseline values.	Continuous, non-invasive, easy to use, and low cost.	Accuracy varies by device.	+\+ Not applicable
3. Blood Pressure (BP).	All ages.	Physiological measure using a digital automatic BP Monitors.	Percentage or absolute increase in blood pressure compared to baseline values.	Standardized and Affordable.	Not continuous, Disruptive during dental care.	+\+ Not applicable
4. Salivary Cortisol Levels.	All ages.	Saliva collection (simulated/unsimulated); ELISA-based analysis.	Quantitative measurement of cortisol levels concentration (ng/mL); compared to baseline or normative diurnal ranges.	Non-invasive, Research-accepted.	Moderate to high cost and takes time to analyze	−\− Not applicable
5. Electrodermal Activity (EDA).	All ages.	Physiological measure using an EDA sensor.	Higher reading indicates dental anxiety.	Continuous and non-invasive.	Moderate to high cost and may cause discomfort.	−\− Not applicable

^ The strengths and limitations are according to PRO. +\+ Presence of validity and reliability. −\− No validity and reliability. MDP: Missing data plan (Did not plan for missing data), * Reported Outcome Measures (PRO).

**Table 11 healthcare-13-02597-t011:** Validity and reliability of DFA tools.

Dental/Fear Anxiety Tools	Validity and Reliability Score #
1. Corah Dental Anxiety Scale (C-DAS).	C-DAS showed a stability score of 0.77 when tested twice within one week. Validity was supported by significant correlations with other anxiety measures, including the Dental Anxiety Question (r = 0.77) and the State-Trait Anxiety Inventory for Children (r = 0.40–0.58, *p* < 0.01) [26].
2. Modified Dental Anxiety Scale (MDAS).	MDAS demonstrated internal consistency ranging from α = 0.799–0.875 in children. Criterion validity was supported by correlations with the CFSS-DS and a single fear question (r = 0.44–0.58). Construct validity was shown, as children reported lower anxiety during orthodontic treatment compared with higher anxiety when treatment involved local anesthesia [31,32,33]. Validated in Arabic, Spanish, Greek, Chinese, Turkish, Romanian, and Tamil [34,35,36,37,38,39,40,41].
3. Modified Child Dental Anxiety Scale (MCDAS).	MCDAS showed internal consistency (item–total r = 0.60–0.74; α comparable to C-DAS and CFSS-DS), with high test–retest reliability over one week. Concurrent validity was supported by correlations with C-DAS and DFSS-SF (r > 0.70), while construct validity was shown by higher scores in girls and older children. Validated for ages 8 to 15 and is reliable for children [42,43]. Also validated in Italian and Nepali [44,45].
4. Faces version of the Modified Child Dental Anxiety Scale (MCDASf).	MCDASf showed internal consistency (α = 0.82) and test–retest reliability (ICC = 0.80). Criterion validity was supported by correlation with CFSS-DS (r = 0.80), and construct validity by higher scores in children with dental anxiety, caries, or requiring general anesthesia. Sensitivity and specificity were 51% and 79%. Validated for ages 5 to 12 and is reliable for children [18]. Also validated in Arabic, Turkish, Croatian, Iranian, Malay, and Chinese [46,47,48,49,50,51].
5. Venham Picture Scale (VPS).	VPS showed internal consistency (KR-20 = 0.838), test–retest reliability (r = 0.70), construct validity through inverse correlation with age (r = −0.47), and concurrent validity through correlation with the Facial Image Scale (r = 0.70, *p* < 0.001). Validated for ages 3 to 18 and is reliable for children [19,53].
6. Children’s Fear Survey Schedule-Dental Subscale (CFSS-DS).	CFSS-DS showed high test–retest reliability (r = 0.97), criterion validity with the Kisling and Krebs behavior rating scale (r = −0.62), and construct validity with higher scores in fearful children (m = 37.8–38.1) compared to non-fearful children (m = 20.9–25.0). Validated for ages 4–14 and is reliable for children [54,55]. Also validated in Arabic, Chinese, Dutch, Finnish, Japanese, Greek, Hindi, Bosnian, Italian, and Swedish [45,55,56,57,58,59,60,61,62,63].
7. Dental Fear Schedule Subscale-Short Form (DFSS-SF).	DFSS-SF showed internal consistency (α = 0.82) and test–retest reliability (r = 0.73). Criterion validity was supported by correlation with the Frankl Behavior Rating Scale (r = −0.54), with higher scores in anxious children (mean 23) compared to non-anxious children (mean 18.7). Validated for ages 8 to 13 years and is reliable for children [67].
8. Smiley Faces Program (SFP).	SFP showed internal consistency (α = 0.80) and test–retest reliability (r = 0.80). Concurrent validity was supported by correlations with the CFSS-DS (r = 0.60) and the MCDAS (r = 0.60). Construct validity was indicated by higher anxiety ratings for invasive procedures (drill and local anesthetic) compared with waiting room and pre-visit situations. Validated for ages 6 to 15 years and is reliable for children [65,68].
9. Revised Smiley Faces Program (SFP-R).	SFP-R showed internal consistency (α = 0.70) and test–retest reliability (r = 0.80). Concurrent validity was supported by correlation with the MCDAS (r = 0.60), and construct validity was indicated by higher scores in girls and older children. Validated for ages 4 to 11 and is reliable for children [69].
10. Facial Image Scale (FIS).	FIS showed concurrent validity through correlation with the VPS (r = 0.70, *p* < 0.001). Validated for ages 3 to 18 and is reliable for children [53].
11. Abeer Children Dental Anxiety Scale (ACDAS).	ACDAS demonstrated internal consistency (α = 0.90) and test–retest reliability (κ = 0.88–0.90). Concurrent validity was shown by correlation with the CFSS-DS (r = 0.77), with a cutoff score > 26 indicating anxiety (sensitivity = 96%, specificity = 66%). Validated for ages 6 and older and is reliable for children. Also validated in Arabic, Turkish, Malay, and Spanish [71,72,73,74].
12. Children’s Experiences of Dental Anxiety Measure (CEDAM).	CEDAM-14 demonstrated internal consistency (α = 0.88), test–retest reliability (ICC = 0.98), and concurrent validity with the MCDAS (Spearman’s r = 0.67). Validated for ages 9 to 16 years and is reliable for children [75]. Also validated in Portuguese and Iranian populations [76,77].
13. Shortened Children’s Experiences of Dental Anxiety Measure (CEDAM-8).	CEDAM-8 demonstrated internal consistency (α = 0.86) with no floor or ceiling effects, strong concurrent validity with CEDAM-14 (r = 0.90), and construct validity with a global dental anxiety item (r = 0.77). Validated for ages 9 to 16 years and is reliable for children [78].
14. Venham’s clinical Anxiety Rating Scale (VCARS).	VCARS showed high inter-rater reliability (r = 0.78–0.98) and construct validity through score differences across sequential visits, and by comparison with heart rate and the VPS. Validated for ages 9 to 16 years and is reliable for children [80].
15. Heart Rate Monitoring.	HR monitoring was validated with mean values ranging from 80 to 118 bpm. Concurrent validity was supported by Pearson’s correlation with the Corah Dental Anxiety Scale (r = 0.57) and by correlation with the numeric Visual Facial Anxiety Scale (Spearman’s rs = 0.48). Validated for ages 6 to 12 years and is reliable for children [15,83].
16. Blood Pressure (BP)	BP monitoring was validated with mean systolic BP 125 ± 9.4 mmHg and diastolic BP 79.9 ± 5.7 mmHg. Concurrent validity was supported by correlation between systolic BP and the numeric Visual Facial Anxiety Scale (Spearman’s rs = 0.29), while diastolic BP showed no significant correlation. Validated for ages 8 to 12 years and is reliable for children [15].
17. Salivary Cortisol Levels.	Not validated.
18. Electrodermal Activity (EDA).	Not validated.

α = Cronbach’s alpha, r = correlation coefficient, ICC = Intraclass Correlation Coefficient, KR-20 = Kuder–Richardson Formula 20, *p* = *p*-value, ρ = Spearman’s rho, κ = Cohen’s kappa, rs = Spearman correlation coefficient. # face validity was conducted in all studies.

## Data Availability

No new data was created or analyzed in this study.

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
