# Peer review of "Assessment of Dental Fear and Anxiety Tools for Children: A Review"

_healthcare, 2025, doi:10.3390/healthcare13202597_

Round 1

Reviewer 1 Report (Previous Reviewer 2)

Comments and Suggestions for Authors

I'd like to thank the authors for responding to the previous comments.  I do note the title of the paper has changed from "Validity and Reliability of Dental Fear and Anxiety Tools: A Literature Review" to "Assessment of Dental Fear and Anxiety Tools for Children: A Review."  I am assuming the manuscript being presented is a revised version of "Validity and Reliability of Dental Fear and Anxiety Tools: A Literature Review" and I will review as such.  

The issue of reliability among each tool described is relatively simple and the paper would not benefit from further development of reliability.  The issue of validity is more complex, as each of the described tools must have been compared against something, and the reader of this paper may have a genuine interest in determining what each scale was validated against.  Listing in tabular or parenthetical form the comparison scale/test/questionnaire/etc. used to validate each tool is important, as it helps the reader understand which scales are "gold standards" and which may have been validated with more questionable methods.  Simply reporting "it is valid" is different than stating "it was validated against the CFSS-DS (r = 0.8)."  As this is the intention of the manuscript based on title, this information is important for every tool described as valid.  If the validation approach does not include comparison against a gold standard, or if only declared valid by an expert, this should be reported.  

Table 11 includes some of these data:  for example, "SFP demonstrated concurrent validity through significant correlations with the DFSS (r = 0.6, p < 0.01) and the
MCDAS (r = 0.6, p < 0.01)."  Firstly, the authors should determine if the abbreviation used in the manuscript refers to the CFSS-DS or DFSS-SF (as "DFSS" alone does not appear in the present manuscript).  Secondly, similar information on how each tool was validated is not present.  If there is no criterion validity data for each tool, information on content or construct validity and how that was determined should be stated.  Example from Table 11:  "VPS demonstrated good test-retest reliability (r = 0.70), high internal consistency (KR-20 = 0.838). Validated for ages 3-11 - Reliable for children."  This is meaningfully different than the previous quote from the same table, as it just states the VPS was "validated" for ages 3-11.  How?  Each tool described as "validated" must have been validated through a process.  That process should be identified, even if the process was simple content validity determined by a single expert.  If there are no data nor a stated process of determining validity, I would argue that the tool has not been validated.  This information is important.

The edited manuscript includes a PRO checklist and something suggested by a manuscript on wearable devices.  I wonder if these edits help the manuscript.  Firstly, there are errors of grammar, syntax, and language--using 'first name, last initial' as a reference (Jessica K), "where" instead of "were", "come in at a low coast," etc.  Secondly, Supplemental Table 1 does not seem to add anything valuable beyond the text and tables presented.  I would consider deleting both of these additions.

Finally, a thorough review for English grammar and syntax should be completed for the entire document.  Examples--P22, line 485 is not a complete sentence ("And to a lesser extent, the Venham Picture Scale (VPS)"); variable capital letters in Table 10 "1. Venham's clinical Anxiety Rating Scale (VCARS)," some statements in Table 10 begin with capital letters, others do not; spelling errors i.e. "sutibale," P16 L 406 "Bp" instead of BP, etc. 

As a final comment, I prefer the initial title.  

Comments on the Quality of English Language

As above, requires close proof-reading and editing.

Author Response

Reviewer 2 Report (Previous Reviewer 3)

Comments and Suggestions for Authors

Thank you for the possibility to review the paper. It has been clarified and improved. The main comment to be addressed now is the discussion section. It now mainly repeats the description of different tools, while it should focus on comparisons between the tools and discuss their use in different situation. Also, there can be a discussion of what previously published articles that compared the tools in the same sample.

Comments on the Quality of English Language

Minor mistakes 

Author Response

Reviewer 3 Report (New Reviewer)

Comments and Suggestions for Authors

Dear Authors,

Thank you for submitting the manuscript titled "Assessment of Dental Fear and Anxiety Tools for Children: A Review" offers a comprehensive evaluation of subjective and objective measurement tools used to assess dental fear and anxiety in pediatric dentistry. 

Overall, your manuscript is well-written, valuable and timely contribution that supports improved assessment and management of dental anxiety in children.

To enhance the manuscript, as authors you should consider the following revisions;

  1. Most tools lack guidelines on handling missing data, identified in the review but not extensively discussed for implications or solutions.
  2. Although comprehensive, the review is qualitative; inclusion of meta-analytic techniques might strengthen the conclusions about tool effectiveness and comparability.
  3. While cost and device availability are briefly mentioned for physiological measures, a more detailed discussion on resource constraints in different clinical settings could improve practical insights.
  4. Although some translations are noted, more detailed analysis of cultural appropriateness and cross-cultural validation for diverse populations would add depth.
  5. Minor inconsistencies are noted in table formatting and labeling that, if corrected, could improve readability and professionalism of the manuscript.
  6. Minor grammar: In some places, singular/plural agreement could be checked (e.g., "lack continuity of measuring" could be "lacks continuity of measurement").

Please check the entire manuscript for typo errors, I listed some of them below;

  • Line 22: "preformed" should be "performed."
  • Line 46: "meas-urement" should be "measurement" (hyphen removed).
  • Line 60: "measurements" should be singular "measurement."
  • Line 176: Multiple "d)" options in answer choices; change last "d)" to "e)" to avoid repetition.
  • Line 379: "Genera1 loud crying" should be "General loud crying" (replace number 1 with letter l).
  • Lines 523, 529, 466: "caost" and "coast" should be "cost."
  • Line 483: Add comma after "And" for better flow: "And, to a lesser extent, the Venham Picture Scale (VPS)."
  • Lines 15, 33, 37, 64, 94, 122, 168, 192, 204, 242, 274, 314, 336, 356, 380, 399, 406: Confirm consistent hyphenation or spacing of words like "non-verbal," "self-reported," "well-being," and age ranges (e.g., "8 years old" vs. "8-year-old").
  • Several occurrences of "sutibale" should be "suitable."
  • "asnwer" should be "answer."

Overall, the document is highly polished with mainly minor typos and some inconsistent formatting. Addressing these will improve accuracy and readability of your manuscript significantly. 

Thank you and best regards,

Round 2

Reviewer 1 Report (Previous Reviewer 2)

Comments and Suggestions for Authors

I'd like to thank the authors for these thoughtful responses to the suggestions made.  This is a thorough and helpful addition to the literature base.  My final suggestion is to make the statement on line 528 a complete sentence.

Reviewer 3 Report (New Reviewer)

Comments and Suggestions for Authors

Dear Authors,

Thank you for the revised version of the manuscript.

Best Regards

This manuscript is a resubmission of an earlier submission. The following is a list of the peer review reports and author responses from that submission.

Round 1

Reviewer 1 Report

Comments and Suggestions for Authors

The manuscript gives an overview of a diverse range of measurement tools to asses dental anxiety or fear in children. It is not the first overview of these kind of instruments, however, updates are always welcome. Such an update would need to be one that critically assesses the known instruments, in order to add value. Unfortunately, the present manuscript does not critically appraise the reliability and validity of the instruments or the papers in which these data are presented. The manuscript is a narrative review that gives an overview with citing what other authors wrote about the reliability and validity. The paper would gain a lot of interest, if the papers referred to in this matter were assessed with regard to their risk of bias.

Additional comments:

  1. The title would be more clear if children are added in one way or the other.
  2. This review is in the abstract called a systematic review, but it isn’t. In the introduction and elsewhere it is not called a systematic review. It would however have been of much more value if it was a systematic review about the reliability and validity of the instruments (including a critical appraisal and risk of bias assessment).
  3. With regard to the search terms the following: no MesH terms seem to be used, there might be other relevant synonyms for children, anxiety scale is singular, anxiety tools is plural. Anxiety tool reveals more hits in PubMed, so this might limit the results found.
  4. The results of the search, the number of articles, belongs to the results section as to my opinion.
  5. Material and method section is very short and lacks information about what is done and what is going to be described in the results.
  6. The results start with a description that might fit better in the introduction. What lacks is an clear announcement to introduce the scales that follow in the next paragraphs.
  7. Reference 27 is not the mentioned paper of Aartman 1998, but to Wright. Please check all references carefully. This was the first I detected, but there might be more.
  8. In the description of the scoring of the CDAS on a scale from 4-20, a reference is made to ref 30, which is about the MDAS, which has 5 items and a scale ranging from 5-25.
  9. In general, all the information about the reliability and validity that is given, is citing original authors that it is all okay. No data are given to check it, or to be able to judge for oneself. In addition, the authors of this paper do not critically appraise, as said before.
  10. The validity and reliability referred to in the paragraphs, seem sometimes to be from papers on adults. Data on validity and reliability for children, should not be based on data from adults.
  11. Paragraph 3.1.4 about the MCDAS again refers to ref 30. Is that correct?
  12. In general, the paper lacks a solid structure.

Reviewer 2 Report

Comments and Suggestions for Authors

Thank you for the opportunity to review this manuscript.

Abstract:  This statement in the results section seems either redundant or better belongs in the methods: "Tools were evaluated based on their target age groups, measurement approach, psychometric properties, strengths, and limitations."  

Introduction:  This statement is difficult to understand in the context of this paper...why mental health researchers require understanding of DFA, why "will they," etc; revision would be helpful: "Researchers focusing on children’s mental health will need to understand various DFA measurement tools, their content, validity, and suitability for different populations." 

P2, L61: why are subjective tools "influenced" by the respondent's viewpoint, and not solely determined by it?

Materials and Methods:  Examples of why the majority of tools identified were excluded would be of value.  A list of those excluded would also be valuable.

P3, L80: The word "data" is plural.

Results:  

Generally, the validity and reliability are just presented as "X said it was reliable and valid."  More information on how validity and reliability were assessed would be valuable, particularly validity.  How are these varied tools assessed for validity if there is no established gold standard?  If there is an established gold standard, or at least a commonly used DFA assessment used as a comparison, it would be important to state this information.  

Generally, the information should be presented in a similar way.  Some "Developed by" have full names, some have Last name, First initial, some have Last Name.  For "Tool Definition," some begin with "it is..." or "it's..." or "is." 

3.1.1-VAS:  The Wong-Baker FACES scale is not considered a VAS. Figure 1 demonstrates a pain assessment, not a DFA assessment.

3.1.9-Wong-Baker:  As above, this is a well-known pain scale.  The authors are encouraged to describe how it has been used or adapted as a DFA scale, with attention to if the exact figure used in Figure 3 was used to assess DFA.  Presenting the original scale is less important than presenting any modifications used to assess DFA rather than pain.  

I would like to know if pulse, BP, salivary cortisol, and EDA have been similarly assessed for validity and reliability for assessment of DFA.

Discussion:  P17, line 450: "The validity of subjective tools decreases when used with younger children."  Is this the case with any of the tools described in this paper?  Or are the age brackets documented in the manuscript what was found to be valid?  P17, line 457: this is not a complete sentence. P17, 459-60: "Shortened Variant" is not a proper noun. 

Conclusions: "Combining subjective and objective assessments may enhance diagnostic accuracy" is stated.  If this was found in the literature review, it was not clearly stated in the manuscript.  

Overall:  this is an overall description of DFA assessments used in pediatric dentistry and should be helpful for providers and researchers.  Addressing the comments above will enhance the manuscript.  What is missing from this review is information on which of these are common and which may have been only described a few times in the literature--comparing the CFSS-DS to something which has been used sparingly may inflate the importance of these uncommonly-used DFA assessment tools.  An enhanced description of how validity was determined--which is the subject of the paper based on the title--should direct the reader to to this point.

Reviewer 3 Report

Comments and Suggestions for Authors

With interest I’ve read the paper “Validity and Reliability of Dental Fear and Anxiety Tools: A Literature Review”. The authors have chosen an important topic and assessed  to review different tools existing for assessing Dental fear and anxiety (DFA) in children.

The article is well-written, but there are a number  comments to be addressed.

Generally grammar and spelling should be re-checked throughout the paper. 

The title should reflect study design, i.e. a systematic review, not just «review».

Abstract is well-written.

Materials and Methods

  • Was the story protocol of the review registered? 
  • Why was not PRISMA guidelines used for reporting this systematic review? Consider using it.
  • What were the exact criteria for the exclusion and inclusion of the articles? State them

Results

Please, add article flow chart or at least a text description of the article flow. How many articles were identified in each database? How many were excluded at different stages of the review? How many were analyzed.

The scales themselves could be moved to supplementary material.

The extracted information can be fully presented in the form of a table (Like table 10). Also, the results of validation (exact results for repeatability, external, internal validity, etc.) for different languages could be incorporated in the table.

After a table, a big-picture results could be presented as well as some specific comments regarding the individual scales.

Discussion

Move table 10 to the Results section.

Besides discussing the scales, it is recommended to search for other reviews describing DFA tools (for example, https://pubmed.ncbi.nlm.nih.gov/17935593/) and to state what is novel in your review.

Comments on the Quality of English Language

grammar and spelling should be re-checked throughout the paper. 
